# Environmental Sampling Methods for Detection of *Salmonella* Infections in Laying Hens: A Systematic Review and Meta-Analysis

**DOI:** 10.3390/microorganisms11082100

**Published:** 2023-08-17

**Authors:** Ewa Pacholewicz, Henk J. Wisselink, Miriam G. J. Koene, Marleen van der Most, Jose L. Gonzales

**Affiliations:** 1Department of Epidemiology, Bioinformatics and Animal Models, Wageningen Bioveterinary Research, Wageningen University & Research, P.O. Box 65, 8200 AB Lelystad, The Netherlands; jose.gonzales@wur.nl; 2Department of Bacteriology, Host Pathogen Interaction & Diagnostics Development, Wageningen Bioveterinary Research, Wageningen University & Research, P.O. Box 65, 8200 AB Lelystad, The Netherlands; henk.wisselink@wur.nl (H.J.W.); miriam.koene@wur.nl (M.G.J.K.); marleen.vandermost@wur.nl (M.v.d.M.)

**Keywords:** *Salmonella*, layers, environmental samples, boot swab samples, sample strategy, meta-analysis

## Abstract

Salmonellosis is the second most commonly reported foodborne gastrointestinal infection in humans in the European Union (EU). Most outbreaks are caused by *Salmonella* Enteritidis, present in contaminated food products, particularly in egg and egg products. In recent years, an increase in the prevalence of *Salmonella* in laying hen flocks in the EU has been observed. For the effective control of infection, adequate detection is key. In laying hen flocks, the occurrence of *Salmonella* in the EU is monitored by the culture of environmental samples (dust, faeces, and boot swabs). The performance of sampling procedures described in the literature for the detection of *Salmonella* in laying hens was reviewed. In total, 924 abstracts were screened, resulting in the selection of 87 abstracts and 18 publications for qualitative and quantitative analyses, respectively. Sample sizes and sampling locations of faecal material and dust were variable and poorly described. Microbiological culture methods used to detect *Salmonella* were variably described in the literature and were often incomplete. Overall, the available literature indicates higher sensitivity of environmental versus individual hen matrices and points to differences in sensitivity between environmental matrices. For non-cage housing systems, boot swabs are the preferred samples, while for cage housing systems dust might be a more reliable sample.

## 1. Introduction

Salmonellosis is the second most commonly reported foodborne gastrointestinal infection in humans after campylobacteriosis in the European Union (EU) Member States (MS) and non-MS countries [1]. The majority of the reported foodborne outbreaks of *Salmonella* were caused by *S.* Enteritidis (SE), present in contaminated food products of animal origin, particularly egg and egg products [1]. Laying hens are thus implicated as the leading source of human *Salmonella* infections. Contamination of egg contents with SE mostly occurs through the colonization of ovaries, oviducts, and vaginal tissue before the shell is formed [2]. Faecal soiling may also contaminate egg shells, and the internal contents of the egg may occasionally be contaminated by organisms entering through hairline cracks in the shell [3].

To reduce the number of human *Salmonella* infections, control programmes have been laid down in the EU MS (EC No. 2160/2003). Regarding the egg production sector, the aim of these control programmes is to identify infected poultry flocks and take effective control measures to prevent human consumption of eggs contaminated with *Salmonella* (EC No. 2160/2003). The success of detection depends on adequate sampling procedures combined with a sensitive culture method. Environmental sampling such as collecting faeces, boot swabs and dust from several parts of the poultry house has been reported as a useful, effective and less invasive method compared to sampling individual birds to predict potential infection or colonization of the poultry flocks [4,5,6]. In 2004, the EU started an EU-wide baseline survey (BLS) for *Salmonella* in which environmental sampling formed the basis [7]. In 2008, *Salmonella* National Control Programmes (NCPs) were implemented in the EU (Regulation (EC) No 2160/2003). Following their implementation, a decreasing trend in *Salmonella* prevalence in laying hens was observed. Since 2014, however, some MS started to report an increase in *Salmonella* prevalence in laying hens [7]. Additionally, the decreasing trend in the number of human salmonellosis cases has levelled off in recent years. There was no statistically significant increase or decrease in the cases from 2016 to 2020 [1]. The reasons for those changes in trends have been investigated recently in the European project ADONIS (“Assessing Determinants Of the Non-Decreasing Incidence of Salmonella”) (ADONIS—One Health EJP). The study included factors related to public health (i.e., more complete reporting and improvements in the reporting and surveillance of human salmonellosis); primary production (deficiencies in the enforcement of existing control measures and sensitivity of statutory sampling programmes); and genetic variation of the pathogen itself. The outcome of the project suggested that the most relevant potential determinants and options for intervention for the stagnating *Salmonella* trend in Europe are factors associated with primary production [8].

A primary production factor associated with this stagnation may be a potential limitation in the efficacy of detection of infected flocks as a result of the sampling approach used in the surveillance programmes. A systematic literature review was performed to gather information on the diagnostic performance of sampling procedures described in the literature for the detection of *Salmonella* in flocks of laying hens. Further, the review aimed to identify the most reliable procedures that could be recommended for standardised application within monitoring programmes. In addition, data were collected on the implementation of the ISO 6579-2002 [9] and ISO 6579-1:2017 [10] standards to culture *Salmonella* in faeces and other environmental samples as described in the reviewed publications. The method, first published in 2002 [9], was updated in 2017 by adding to its scope samples from the primary production stage [10]. How the standards were followed in the reviewed publications was evaluated.

## 2. Materials and Methods

### 2.1. Research Questions

For this study, the flocks of laying hens were defined as the population of interest. The intervention of interest was the sampling strategy for the detection of *Salmonella* infection at the flock level using environmental samples as well as individual bird samples and the outcome of interest was the diagnostic confirmation of *Salmonella* presence in the flock. Therefore, the following questions were defined: (Q1) Which environmental samples were used to determine *Salmonella* status in laying hens? (Q2) How were the individual laying hens sampled to determine the status of the flock? (Q3) What was the diagnostic (relative) sensitivity among different environmental sampling methods (defined by i.e., the type of sample, sample size, etc.) to detect *Salmonella* in laying hens? In addition to these three questions, the collected literature was assessed for (Q4) which steps of the culture methods described under ISO 6579-2002/ISO 6579-1:2017 [9,10] have been applied?

### 2.2. Search Strategy

This Systematic Review followed the Preferred Reporting Items for Systematic Reviews and Meta-Analyses (PRISMA) guidelines, including quality assessment and transparency [11]. The search consisted of four steps: literature search (1), screening and quality assessment (2), data extraction (3) and data analysis and summation (4). The search for relevant publications was performed using two databases: the CAB Abstracts database provided by the Wageningen University and Research Library (Wageningen UR, Wageningen, The Netherlands) and the Scopus database (Elsevier, Amsterdam, The Netherlands).

To find keywords related to environmental sampling methods for the detection of *Salmonella* infections in laying hens a small subset of relevant publications was reviewed. The identified keywords, as indicated in Table 1, were entered into the two databases. Searches in CAB abstracts were in the ‘multi-purpose’ (.mp.) fields for this database: abstract, title, original title, broad terms, heading words, identifiers and cabicodes. Searches with the keywords in Scopus were in title, abstract and keywords. The databases searched automatically both the single and plural forms of a keyword without adding any special code. To increase the search power, an asterisk (*) was used at the end of the root of a word to instruct the databases to search for all forms of a word, e.g., prevalen* retrieved prevalence or prevalent. Boolean operators AND and OR were used to narrow down the search. No restrictions were imposed on the publication date or geographical location. The search language was in English and the last search was performed on 12 January 2022. Retrieved records were imported into the online tool CADIMA [12] and checked in this tool for duplicates.

### 2.3. Inclusion Criteria and Records Screening

The search strategy was validated for reliability, using a subset of publications already identified as relevant to the objective. Studies retrieved from the databases were assessed against the inclusion criteria for relevance and eligibility. The criteria were as follows:The study is performed in laying hens;*Salmonella* is the subject being studied;Type of sampling in the environment is described;Detection (or not) of *Salmonella* in the environment is described;The prevalence of *Salmonella* in the flock is described;The publication is written in English;The study is peer-reviewed;The study described is primary research;The full text of the study is available;Data are available on individual farm level (used to discriminate publications for the qualitative or quantitative data extraction).

The screening was performed in two stages. First, a title and abstract screening were performed for relevance and eligibility followed by a full-text screening. If no abstract was available or any of the inclusion criteria mentioned above could not be properly evaluated based on the title and abstract alone, the eligibility of those studies was evaluated based on the full text. By using the online tool CADIMA, the title/abstract screening was performed by one reviewer. If this reviewer considered a record relevant or unclear, it was included in the full-text screening, and if the record was not considered relevant, the record was placed on the list of non-relevant records. Ten per cent of the publications were randomly selected in CADIMA and screened by a second reviewer. To accept title/abstract screening by the first reviewer as valid, the agreement in results of the screening of titles and abstracts between the two reviewers should be above 90%. In this study, it was 98%. To perform the full-text screening, the manuscripts identified in the title/abstract screening were uploaded in CADIMA and the screening was managed by two different researchers simultaneously. A PRISMA flowchart was used to summarize all stages of the manuscript selection process [11].

### 2.4. Data Extraction

Qualitative and quantitative data were extracted from all eligible manuscripts. Qualitative data included: (1) the characteristics of the included studies: reference, i.e., CADIMA ID, published year, name of first author, country of isolation; (2) the characteristics of the farm: housing system; (3) the details of the environmental sample methods: sample matrix (type of sample), size of the samples, location where the samples were collected; (4) the characteristics of the sampling of individual laying hens: sample matrix; (5) details of the *Salmonella* culture methods used in the different steps of the ISO 6579-2002/ISO-6579-1:2017 [9,10] method: pre-enrichment, selective enrichment, plating out agars, biochemical testing, serological testing. Quantitative data included: (1) the characteristics of the included studies: reference, i.e., CADIMA ID, reference to the tables in the publications from which the data were extracted, name of first author and publication year, country of isolation; (2) the characteristics of the farm: ID of the flock/farm (some studies examined multiple flocks per farm, whereas others single flocks from multiple farms), housing system; (3) results of the testing of the individual laying hens for *Salmonella*: number tested, number positive and sampling matrix (4) results of the testing of environmental sampling for *Salmonella*: number of samples tested, number of positive samples, the details of the environmental sample methods (i.e., sampling matrix and location) and *Salmonella* serotypes.

### 2.5. Data Analysis

The data analysis included both qualitative and quantitative analyses.

#### 2.5.1. Qualitative Analysis

In the qualitative analysis, the environmental matrices used for the detection of *Salmonella* in laying hens were summarized, including the type of matrices, size of collected samples and their collection location. In addition, data on matrices used to sample individual hens were summarized. Results were visualized using the statistical software R version 4.2.2 [13]. Furthermore, the analysis summarized which steps of ISO 6579-2002/ISO 6579-1:2017 [9,10] were described for the detection of *Salmonella* in environmental samples.

#### 2.5.2. Quantitative Analysis

The extracted quantitative data were analysed to compare the proportion of positive samples obtained by different environmental sampling matrices whilst controlling for the effect of variables such as within flock prevalence (proportion of positive hens as determined by individual sampling of birds). In short, a Generalized Linear Mixed-Effect Model (GLMM) was fitted with a binomial distribution where the response variable was the proportion of positive environmental samples. The explanatory variables (fixed effects) included were the type of environmental sampling matrix, the proportion of positive hens in the flock (used to adjust for within flock apparent prevalence), the housing system and the interaction of the environmental sampling matrix and housing system. The model included the random effect on flock/farm nested within a publication and was corrected for cluster effects (autocorrelation) due to observations originating from the same flock/farm and publication. The model was fitted in the statistical software R [13], using the library lme4 [14]

## 3. Results

### 3.1. Literature Search and Screening

Figure 1 summarizes the search and screening procedures. Through the database search, there were 1241 publications identified. No additional records were identified through sources other than those indicated in the methods. After the removal of duplicates, 949 records remained for the screening of abstracts and titles. There were 835 excluded publications and 114 continued to the full-text screening by both reviewers. Here, 27 were deemed ineligible due to the lack of environmental sampling or original data or to being unrelated to laying hens. As a result, 87 publications were eligible for qualitative analysis and 18 for quantitative analysis. The difference between qualitative and quantitative eligibility was related to the availability of raw data. Many publications provided aggregated data, disabling their inclusion in the meta-analysis; however, they provided valuable input for the qualitative extraction. Additionally, the 87 selected publications were screened for details about the culture methods for *Salmonella* described under ISO 6579-2002/ISO 6579-1:2017 (Q4) [9,10]. In seven publications, no details were mentioned about culture methods. For this reason, these publications were excluded for further analysis and extraction of data was performed on 80 publications. A list of eligible publications to answer the particular questions is included in Appendix A.

### 3.2. Qualitative Analysis

The timeline of publishing of the 87 publications extracted for qualitative analysis covered the years from 1991 to 2021, as depicted in Figure 2A. The eligible sources for the qualitative assessment originated mostly from the US (26%), the UK (16%) and Australia (10%) (Figure 2B).

#### 3.2.1. Overview of Environmental Sampling Matrices (Research Question One)

Table 2 summarises the environmental sample matrices described in the literature included in this review, consisting of types of matrices grouped in sources of samples such as faecal material, dust, nest boxes, feed, water, poultry house equipment, pest, poultry house area, environmental samples unspecified and other source of samples. Each of these groups also contains subgroups, which are listed in Appendix A.

Faecal material and dust were the most frequently described environmental matrices in the literature for the detection of *Salmonella* in laying hens. However, the sampling method was highly variable as depicted in Figure 3. The definition of faecal material samples was variably interpreted. The types of matrices described in the publications and grouped into the category “faecal material” included pooled faeces, litter, boot swabs/overshoes from litter, swabs from faeces/litter, swabs/sponge from manure belt/scraper and sponge from faeces. Two publications reported the collection of boot swabs outside of the houses [17,18]. Studies reported pooled samples, varying in the size of the pool, i.e., from a few droppings up to samples of more than 200 g. There were three publications in which sampling of litter (bedding) in grams was described [19,20,21]. In addition, the location of sampling was variable and generally poorly described. It was indicated that a sample was collected through the pen/house/shed, or from cages, rows or belts, floor or floor litter. Specific locations, where the samples were collected, e.g., within a pen, and the quantity of samples collected were not specified, making it difficult to reproduce the procedure described in particular studies.

Similarly to faecal material samples, dust sampling was highly variable as well (Figure 3B). In this review, samples grouped into the category “dust” included pooled samples of dust, swabs, drag swabs, sponge swabs, mixed samples, shoe covers, and nurse cups taken from surfaces not in contact directly with faecal droppings (e.g., litter). There were variable names given to some types of samples described in the publications, although the meaning might have been the same, i.e., it was assumed that boot swabs, shoe covers and nurse cups indicate the same type of sample. These matrices were included in the category “Dust” only in cases where it was explicitly indicated in the publications that a dust sample was taken that way. In some cases, especially for the cage housing system, it was not clear what matrices were sampled using boot swabs and it was assumed that boot swabs were taken whilst walking along corridors between cages.

There was no uniform way of collecting the dust sample and frequently it was not specified how the sample was collected. The quantity of the samples varied from <50 g up to >200 g, or 250–500 mL. The location of the sampling was variable and poorly described. The descriptions of locations included cages, belts, through the house/shed/pen, from fans, walls, nest boxes, ledges, beams and pipes. The sampling locations were also dependent on the housing system. According to one study in a cage housing system, dust was collected from the floor beneath the cages and egg elevators. In non-cage systems, dust was collected from air exhaust baffles, surfaces of nest boxes, ledges and horizontal beams [5].

Out of 87 publications eligible for qualitative analysis, the housing system (grouped into cage and non-cage) was reported in 65 studies. The category “cage” housing system included systems described in variable ways such as conventional cage, cage not specified further, battery cage, battery cage automatic/manual, belt cage, scraper cage, colony cage, enriched cage, enriched colony cage and step cage. The category “non-cage” included housing systems described as aviary, barn, cage free, floor, floor pens, floor raised, free range, free-range floor, on floor, open house, organic and outdoor.

#### 3.2.2. Overview of Matrices for Sampling Individual Hens (Research Question Two)

The sampling of individual hens was described in 51 of the publications eligible for qualitative data extraction. Table 3 summarises samples of hens and a number of publications where particular samples were described. Most frequently sampled were eggs. Eggs, either when using internal content or shells for diagnosis, were considered matrices for sampling individual hens. After eggs, the most frequently described samples were cloacal swabs and caeca. One study reported the sampling of ovarian follicles from dead layer birds [22], and another one reported on yolk from dead birds [23]. These samples were counted as “ovary” samples”. Two studies reported sampling organs of spent hens, these were counted in the “organs category” [20,24]. Two publications described sampling individual droppings, which were considered as samples from individual hens [5,6]. In 27 publications, the hens were sampled using one matrix, mostly eggs (21 publications) or cloacal swabs (six publications). In the remaining 24 publications, multiple types of matrices were sampled, of which the most frequent was the combination of eggs and cloacal swabs (four publications).

### 3.3. Quantitative Analysis

#### 3.3.1. Detection of Positive Flocks Based on Sampling Individual Birds versus Environmental Matrices

Out of 18 publications used to extract data for the quantitative analysis, 13 provided data on the apparent prevalence in hens based on any of the following samples taken from individual birds: eggshells [25,26,27,28], egg contents [19], cloacal swabs [15] and individual droppings [5]. In two publications [4,29], multiple organs, ovaries, oviduct and caeca were taken per bird. Here, a bird was considered positive for *Salmonella* if at least one of these samples tested positive. In one publication, in addition to the ovaries/oviduct/caeca collected, there were also faecal samples collected from individual birds [6]. From that study, the results from ovaries/oviduct/caeca were extracted, since the percentage of positive birds in that study was higher when testing those matrices. Multiple matrices from individual birds were described in four additional publications. From these publications, the matrix having the highest percentage of *Salmonella*-positive samples was extracted. One publication described matrices as liver, spleen, ovary, caecum and cloacae. Data from the latter matrix was extracted [30]. From another publication, data on caeca was extracted, although cloacal swabs were also described [31]. In another publication [28], the egg shells and floor eggs were reported, of which data on eggshells only were extracted. One publication [20] reported sampling organs and caeca from spent hens. To enable statistical analysis and use the estimates of each flock’s apparent prevalence (computed using different sample matrices), it was assumed that the sensitivity for detection of *Salmonella* of these different matrices was similar. The observed apparent prevalence at the flock level had a median of 3.3% (first quartile 0%–third quartile 18.0%). The summary of environmental sampling matrices extracted from the eligible 18 publications and the reported results are summarised in Table 4. The table provides a descriptive summary of farms/flocks positive for *Salmonella* depending on the sampled matrix.

A comparison of the detection of positive flocks/farms based on sampling environmental matrices and individual hens (based on 13 publications) is provided in Table 5. It shows that the differences in the number of positive flocks detected by both approaches (environmental samples and individual hens) are minor. However, the number of samples taken from individual hens as compared to the number of environmental samples was higher. Thus, there is a marked difference in the effort needed to either sample and test individual birds or environmental samples to reach a similar sensitivity of detection

#### 3.3.2. Comparison of Proportion of Positive Samples between Environmental Matrices (Meta-Analysis—Research Question Three)

The meta-analysis, i.e., the relative comparison of the proportion of positive samples between matrices, was based on eight publications [4,15,19,20,25,26,27,29] including 378 observations. The results from a multivariate mixed effect model, adjusted for within flock apparent prevalence are presented in Appendix A and summarised in Figure 4 below. The results revealed that in non-cage housing systems, boot swabs had a higher probability of detection compared to pooled faeces (Odds = 1.4 times higher, *p* = 0.08) and dust (Odds = 1.8 times higher, *p* = 0.001); however, the difference with dust matrix was statistically significant only if considering *p* < 0.05 as a threshold for significance. In cage housing systems, the highest mean probability of detection was found when sampling boot swabs. This probability did not differ significantly from the other matrices. When comparing sample matrices within the caged system, dust had a significantly higher probability of detection than faeces (Odds = 1.3 times higher, *p* = 0.002) (Appendix A).

The model confirmed that the detection of positive samples, using any of the environmental matrices, increased with an increase in the apparent prevalence in hens (*p* < 0.001) (Appendix A).

### 3.4. Use of ISO 6579-2002/ISO 6579-1:2017 for Detection of Salmonella (Research Question Four)

Details of culture methods were described in 80 full texts used for qualitative analyses, including studies from 27 countries published between 1991 and 2021. In 21 (26%) and one (1%) studies it was mentioned that culture methods were performed in accordance with ISO 6579-2002 [9] and ISO 6579-1:2017 [10], respectively. Of these, the first was from 2008, with 17 studies from the period 2008–2021.

One study from 1995 was performed in line with the ISO 6579-2002 [9] procedure, although it was published before the first ISO 6579-2002 procedure was published in 2002. In more detail, in 65 (81%) studies, buffered peptone water (BPW) was used for pre-enrichment, as described in the ISO procedure. Next, for selective enrichment of *Salmonella* spp., MSRV or RVS was used in 75 (94%) of the studies. For selective plating-out, *Xylose Lysine Deoxycholate agar* (XLD; Oxoid, Basingstoke, Hampshire, UK) was used in 34 (43%) of the studies either alone (*n* = 4) or in combination with one or more second selective solid media (*n* = 30). As a second selective solid medium to XLD, most frequently Brilliant Green Agar (Oxoid, Basingstoke, Hampshire, UK) was used (*n* = 11), followed by MacConkey Agar (Oxoid, Basingstoke, Hampshire, UK; n = 7), Brilliance^TM^ Salmonella Agar (Oxoid, Basingstoke, Hampshire, UK; n = 6), Rambach^TM^ Agar (Merck-Millipore, Burlington, Massachusetts, US; *n* = 5), Hektoen Enteric Agar (Oxoid, Basingstoke, Hampshire, UK; *n* = 5) and one or two times in five other selective solid media. XLD was not used in 46 (57%) of the studies. In 42 of these studies, one or more other selective solid media as XLD were used whereas in four studies no use of such a medium was reported. In total, the use of 17 selective solid media was described in 76 (95%) studies. A complete breakdown of these media is in supplemental Appendix A. Biochemical and serological confirmation of the isolated *Salmonella* bacteria was described in 60 (75%) and 71 (89%) of the studies, respectively. Overall, in 18 (23%) of the studies, the described methods matched the complete ISO 6579-1:2017 [10] procedure (Figure 5).

## 4. Discussion

This review summarized the sampling procedures described in the literature for the detection of *Salmonella* in laying hen flocks. The sampling procedures were summarized both for environmental samples (Q1: Which environmental samples were used to determine *Salmonella* status in laying hens?) and individual laying hens (Q2: How were the individual laying hens sampled to determine the status of the flock?). This was followed by a meta-analysis on the diagnostic performance of the sampling procedures using eligible sources (Q3: What was the diagnostic (relative) sensitivity among different environmental sampling methods to detect *Salmonella* in laying hens?). In addition, it summarized how the ISO 6579-2002/ISO 6579-1:2017 [9,10] standards were followed in the reviewed publications (Q4: which steps of the culture methods described under ISO 6579-2002/ISO 6579-1:2017 have been applied?).

The results from the qualitative analysis of matrices for sampling flocks of laying hens revealed that faecal material and dust were the most commonly tested environmental matrices (Table 2), whereas eggs were the most common matrix to sample individual birds (Table 3). Eggs were chosen as matrices to sample individual hens. However, when testing eggshells, one cannot exclude that a positive result is due to environmental contamination rather than infection of the individual hen. The separation between environmental samples and individual samples of laying hens was not always clear in the publications reviewed. In general, there was a large heterogeneity in sampling procedures for *Salmonella* detection in environmental samples and individual hens as described in the literature (Q1, Q2). There were variations in sampling protocols describing the sampling matrices, number of samples and sampling locations. All these variables influence the efficacy of detection [4,5].

The summary of the percentage of detection of positive flocks/farms based on sampling environmental matrices and individual hens (Table 5) indicated that differences in the number of positive flocks detected by both approaches were minor when the aim was detection of infected flocks/farms. Nevertheless, the environmental samples might be favourable since, given the lower number of samples taken, they appear to be more cost effective than sampling individual birds. The obtained results confirm those from a previous report [4].

The quantitative meta-analysis (Q3) was based on a relative comparison of environmental sampling matrices between each other, whilst controlling for the effect of the apparent within-flock prevalence (as measured using different sample matrices from individual birds). As expected, the results showed that for all environmental matrices, a higher prevalence in hens increased the probability of detection of a *Salmonella*-infected flock (Appendix A), as previously demonstrated [4]. The main aim of this study was to identify the sampling matrix most likely to detect *Salmonella* infection in a flock/farm. The meta-analysis results showed that for non-cage housing systems, boot swab sampling appears to provide the highest probability of detection, with the odds for detection being 1.4 and 1.8 times higher than pooled faeces and dust, respectively. From these matrices, dust sampling in non-cage systems had the lowest sensitivity, with the probability of detection being also lower than pooled faeces. For caged systems, on average a higher probability of detections can be expected when using boot swabs compared with other matrices. However, due to a limited number of observations, there is a large uncertainty (broad confidence intervals) (Figure 4) around this estimate that limits statistical confirmation of the findings. What could be statistically confirmed is that for caged systems, dust samples were 1.3 times (odds for detection) more sensitive than pooled faeces. The latter showed the lowest probability of detection. Different sensitivities of sampling matrices depending on the housing system, as observed for dust and faeces samples, were also reported earlier [29,32]. The reasons for these differences are unclear. The median number of dust and pooled faeces samples taken per flock/farm for both caged and non-caged systems was similar (Table 5). Hence, it was assumed that sample size may not contribute to the observed statistical differences in the analysis. A potential explanation of this difference could be that faecal material from infected birds is more homogeneously distributed through a non-cage facility, since the birds may move through it. In contrast, the hens in a cage system do not move through the house. This limits the detection of positive faeces to the cages where infected hens are present and have been selected for sampling.

The review gives insight into how the culture method according to ISO 6579-2002/ISO 6579-1:2017 [9,10] was described in the reviewed publications. In 2017, the method was updated by including sampling from primary production and the option to choose between using the broth (RVS) or the semi-solid agar of Rappaport-Vassiliadis medium (MSRV). While reviewing the publications included in this study, it was found that the culture method was often incompletely described (Q4). As detection of *Salmonella* is not only affected by the sampling procedure but also by the culture method, the results of the data extraction were added to this review. They revealed that different steps of culture methods as described in ISO 6579-2002/ISO 6579-1:2017 [9,10] have been widely applied, but variation from this ISO method was common. In particular, the step of cultivation on XLD plates was often not included. It is difficult to say to what extent these variations affected the final result for the detection of *Salmonella*. The findings from this review suggest that there is still room for improvement in terms of clearly reporting the materials and methods used in the studies.

During this review, several assumptions were made. The large variability in sampling matrices described in the literature on environmental sampling level led to the aggregation of the extracted data into groups in order to enable data analysis. It was assumed that any individual sample had the same sensitivity and so were treated equally. The meta-analysis combined the results from several publications and aggregated different types of particular matrices into one category. The high variability between the matrices and poor description provided in the publications prevented further distinction in the way the samples were collected, or in the volume (e.g., dust) of the samples. As reported based on field data, the sensitivity of detection can be influenced by the size of the sample (e.g., the size of the faeces pool) [5], as well as by the number and combination of faeces and dust samples [4]. Also, in extracting the data on the prevalence in hens based on individual birds, it was assumed that the samples from different individual hen matrices had the same diagnostic performance. Differences have been reported, however. In one study the rate of contamination of eggshells was higher than for contents [6]. In another study, cloacal swabs had a higher percentage of positive results compared to liver, spleen, ovary and cecum [30]. These limitations bring some level of uncertainty to the statistical findings of this study. However, the results provide a confident indication that environmental sampling is as reliable as individual hen sampling for detecting infected flocks and more cost effective, since fewer samples are required. In addition, extracted data provides an indication that boot swabs may be the sampling matrix of choice for both caged and non-caged housing systems.

The ultimate aim of this review was to identify best practices (sample matrix, the number/volume of samples, sampling location: nests, egg belts, barn, etc.) that can be recommended for standardised application within monitoring programmes. Unfortunately, data were very limited because the methods used were often insufficiently described. When available, there was a wide variation in both quantities of matrices and locations sampled. In conclusion and based on extracted data from the literature, it is difficult to advise on the best environmental procedure to sample laying hens. However, some general recommendations can be provided.
When the purpose is to detect infected flocks, environmental samples, particularly boot swabs would be the recommended samples, since they are more sensitive than those from individual hens and fewer numbers are required.Based on relative comparison in non-cage housing systems, pooled faeces seems to be superior to dust samples. For caged systems, dust gave better results. However, since the EU plans to phase out caged animal farming, this may be less relevant in the future. The use of pooled faeces is recommended for non-caged systems if samples other than, or in addition to, boot swabs are to be collected.The limited data available from the literature, unfortunately, does not allow for making recommendations on the best sample size or sample locations within the laying hen house. It is recommended to gain new experimental data to address this issue.The findings from this review suggest that there is room for improvement regarding reporting of the methods used. It is recommended to use uniform terminology in naming matrices used for sampling laying hens.

## Figures and Tables

**Figure 1 microorganisms-11-02100-f001:**
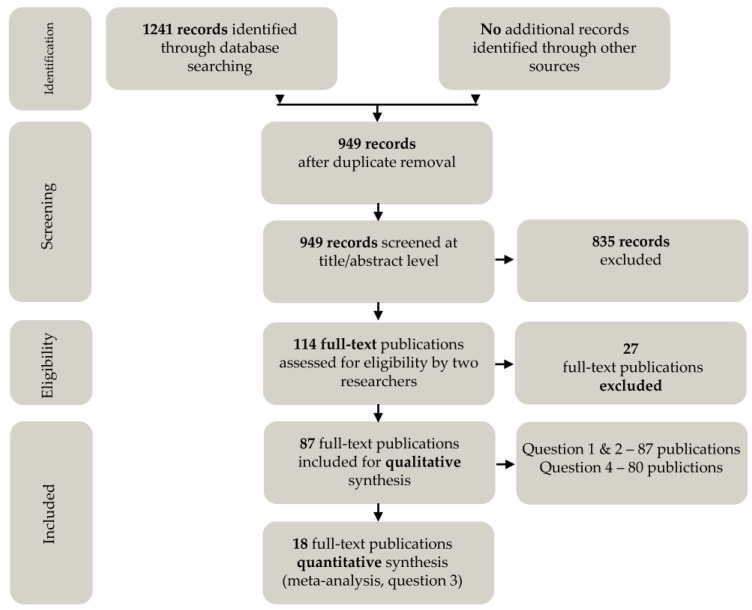
The number of the included and excluded publications during the screening of the databases, screening of titles/abstracts and full texts. There were 87 full-text articles eligible for qualitative synthesis to answer Q1 “Which environmental samples were used to determine *Salmonella* status in laying hens?” and Q2 “How were the individual laying hens sampled to determine the status of the flock?” Of these 87 publications, 80 were eligible to answer Q4 “which steps of the culture methods described under ISO 6579-2002/ISO 6579-1:2017 [9,10] have been applied?” Further, 18 publications were eligible for quantitative synthesis to answer Q3 “What was the diagnostic (relative) sensitivity among different environmental sampling methods to detect *Salmonella* in laying hens?”.

**Figure 2 microorganisms-11-02100-f002:**
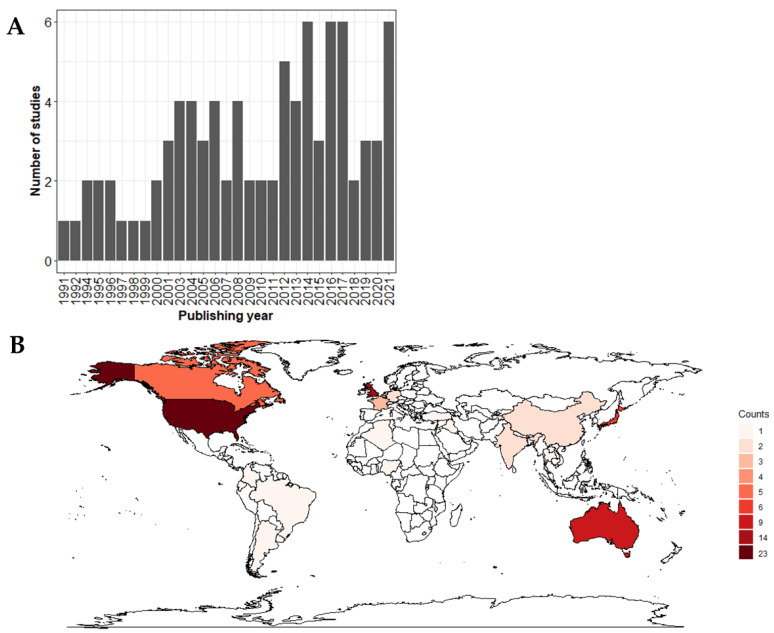
Number of extracted publications for qualitative assessment by year of publication (**A**). Number of extracted publications for qualitative assessment by country where the samples were collected (**B**). Two publications described data from more countries [15,16].

**Figure 3 microorganisms-11-02100-f003:**
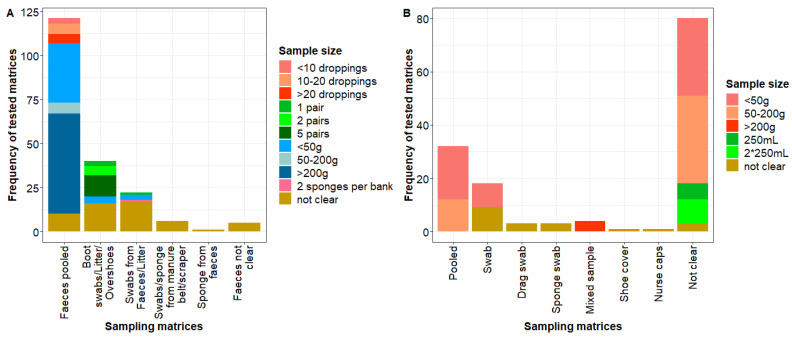
Variability of sampling faecal material (**A**) and dust (**B**) matrices as described in the literature. (**A**) summarises data extracted from 69 publications, whereas (**B**) summarises data from 43 publications. On the X-axis, different types of samples as described in the literature are listed. The Y-axis summarises the total number of observations describing a particular matrix. Matrices could be described multiple times in a single publication.

**Figure 4 microorganisms-11-02100-f004:**
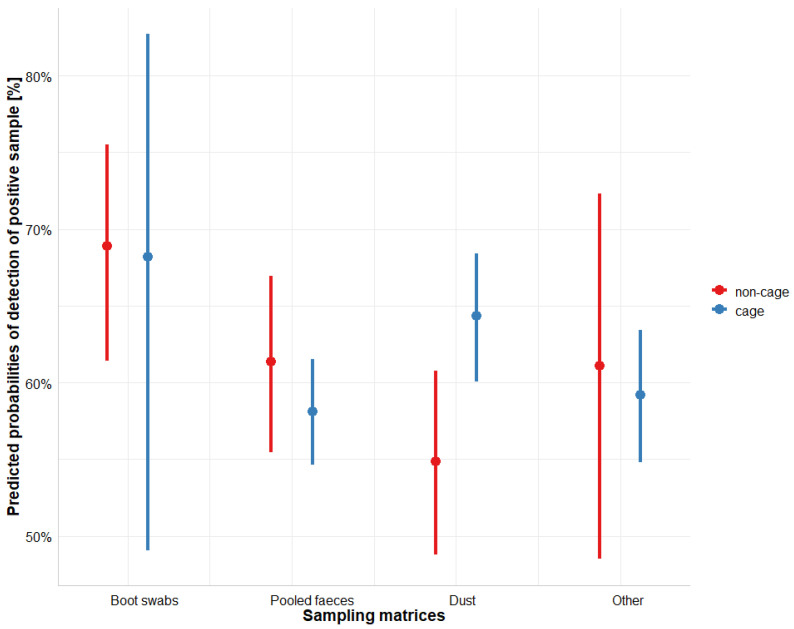
Results of the logistic regression model to compare the proportion of *Salmonella* positive samples depending on the sampling matrix, with boot swabs samples in a non-cage housing system as reference. Data originate from eight publications. Summary of the logistic regression model is included in Appendix A.

**Figure 5 microorganisms-11-02100-f005:**
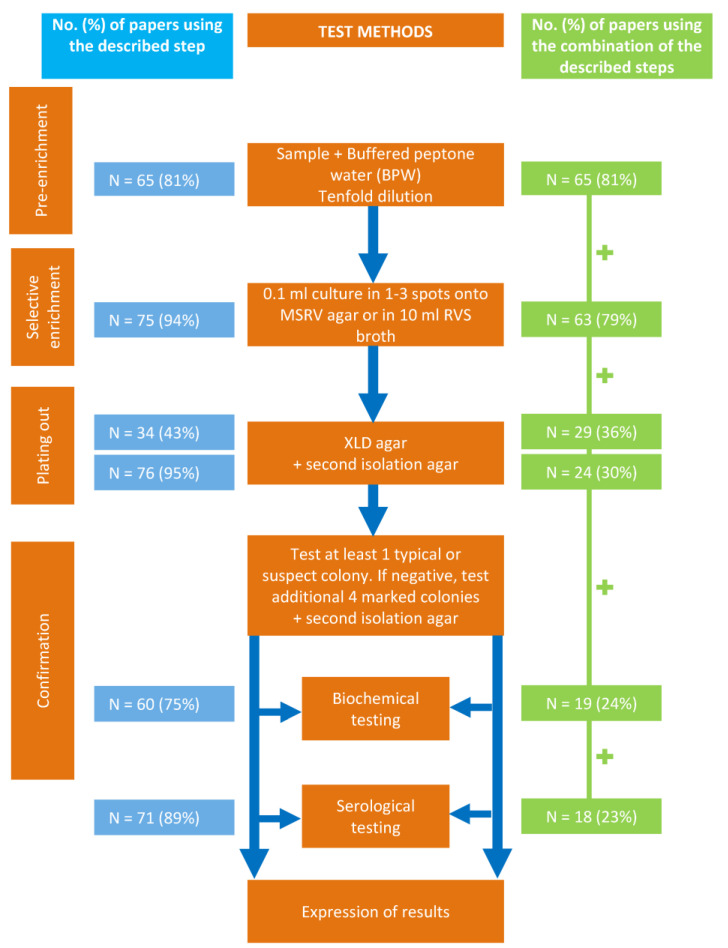
Diagram of the ISO 6579-2002/ISO 6579-1:2017 [9,10] procedure for detection of *Salmonella* in animal faeces and in environmental samples from the primary production stage. The different culture steps were indicated in orange colour. The results of the systematic literature review were indicated in blue and green colour and showed the number of studies which reported the specific culture step (blue colour) and the combination of the culture steps (green colour).

**Table 1 microorganisms-11-02100-t001:** Search strategy in literature databases.

Search Nr.	Keywords	No. Publications
CAB Abstracts on Ovid Platform		
1	Salmonella.mp.	68,799
2	(layer* or laying or egg-laying).mp.	263,394
3	(environment* or prevalen* or monitor* or surveillance or boot? or swab*).mp.	1,855,687
4	1 and 2 and 3	734
Scopus		
1	(TITLE-ABS-KEY (salmonella)) AND (TITLE-ABS-KEY (layer? OR laying OR egg-laying)) AND (TITLE-ABS-KEY (environment* OR prevalen* OR surveillance OR monitor* OR boot? OR swab?))	444

**Table 2 microorganisms-11-02100-t002:** Overview of the environmental sampling matrices reported in the eligible publications (data extracted from 87 publications) to detect *Salmonella* in laying hens.

Source of Samples	Number of Publications
Faecal material	69
Dust	43
Nest boxes	15
Feed	26
Water	22
Poultry house equipment	37
Pest	21
Poultry house areas	21
Environmental unspecified	15
Other	6

**Table 3 microorganisms-11-02100-t003:** Overview of the samples as reported in the literature that were taken from individual laying hens to detect *Salmonella* and number of publications where particular samples were described. Data were extracted from a total of 51 publications in which sampling of the hens was described.

Type of Matrix		Number of Publications	
Eggs		34	
Part(s) of an egg used for diagnosis	Shells		18
	Contents		19
	Whole		2
	Non-specified		8
Cloacal swabs		15	
Intestinal tract		15	
Caeca			13
Intestines			3
Organs		12	
	Liver		8
	Spleen		10
	Heart		3
	Gallbladder		1
Reproductive tract		13	
	Ovary		13
	Oviduct		8
	Upper reproductive tract		1
	Uterus		1
Individual faecal droppings		2	

**Table 4 microorganisms-11-02100-t004:** Overview of positive farms/flocks based on sampling of environmental matrices extracted from a total of 18 publications providing quantitative data. This summary includes number of publications reporting on particular matrices, number of flocks/farms positive, number of tested farms/flocks and the percentage of positive flocks/farms.

Housing System	Environmental Sampling Matrices	Number of Publications	Flocks/Farms Positive	Flock/Farms Tested	Percentage Positive Flocks/Farms
Cage	Pooled faeces	9	117	175	67
Cage	Boot swabs	2	5	5	100
Cage	Dust	7	91	123	74
Cage	Other	8	93	111	84
Non-cage	Pooled faeces	5	39	63	62
Non-cage	Boot swabs	5	35	92	38
Non-cage	Dust	4	39	83	47
Non-cage	Nest box	2	13	35	37
Non-cage	Other	3	24	62	39
Non-specified	Pooled faeces	4	64	119	54
Non-specified	Pooled faeces or Boot swabs	1	17	20	85
Non-specified	Boot swabs	1	1	1	100
Non-specified	Dust	3	61	99	62
Non-specified	Other	2	5	41	12

**Table 5 microorganisms-11-02100-t005:** Comparison of positive flocks/farms based on sampling environmental matrices or individual hens. Results are obtained from 13 publications which described quantitative data on both environmental samples and samples from individual hens (type of matrix is not specified in the table, but overall summarised under Section 3.3.1). Multiple environmental samples were described in publications.

Environmental Sampling Matrices	Number of Publications	Number of Flocks/Farms Tested	Number of Positive Flocks/Farms Based on Environmental Sampling/(%)	Number of Environmental Samples per Farm/Flock Median (Quartile 1–3)	Number of Positive Flocks/Farms Based on Sampling Hens/(%)	Number of Individual Samples per Farm/Flock Median (Quartile 1–3)
Pooled faeces	10	257	168/65%	10 (5–30)	137/53%	60 (40–100)
Pooled faeces or Boot swabs	1	20	17/85%	10 (5–10)	16/80%	296 (225–300)
Boot swabs	7	66	32/48%	2 (2–6)	34/52%	30 (30–100)
Dust	8	271	165/61%	10 (5–27)	165/61%	100 (30–100)
Nest box	1	28	8/29%	10 (10–10)	4/14%	30 (30–30)
Other	7	86	53/62%	20 (9–21)	23/27%	30 (30–60)

## Data Availability

All relevant data generated or analysed during this study are included in this article and its Appendix A.

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
