# Peer review of "Environmental Sampling Methods for Detection of Salmonella Infections in Laying Hens: A Systematic Review and Meta-Analysis"

_microorganisms, 2023, doi:10.3390/microorganisms11082100_

Round 1

Reviewer 1 Report

The authors present an intriguing manuscript that reviews the available information with the aim of gathering data on the diagnostic performance of sampling procedures described in the literature for detecting Salmonella in flocks of laying hens. Furthermore, the review aims to identify the most reliable procedures that could be recommended for standardized application within monitoring programs. Additionally, data were collected on the implementation of the ISO 6579-2002 and ISO 6579-1:2017 standards for culturing Salmonella in feces and other environmental samples, as described in the reviewed publications. Complementarily, the discussion encompasses all aspects that could limit the interpretation of the results and provides guidance for the development of future research in the field. The article is well-written and easy to follow, and the analysis methods used are appropriate. I only have minor comments on the manuscript before its publication.

In Table 2A, I suggest removing the numbers that are within the figure. As for Figure 2B, I recommend evaluating the presentation of the results in a graphical format (map) that allows for a clearer visualization of the origin of the identified studies.

In the information regarding the different types of matrices reported in the studies for detecting the presence of Salmonella (pages 6-7, and Table 2), is it possible to identify any trend or association between the year of the study and the type of matrix used?

Reviewer 2 Report

Reviews and meta-analyses are always a worthwhile exercise. A review of sampling poultry/poultry environments for Salmonella is needed, and the authors have made a good job of addressing this. Overall, the paper is well-presented and clearly written, with appropriate methodology. It is unfortunate that few firm recommendations could be made, but this is a reflection of the variability of methods etc in the reviewed papers. This in itself is a notable observation that will hopefully make future authors consider their reports more fully.

I only have a few minor points that need addressing (see attached doc). The main point for consideration is in the use of alternatives to XLD. More information on the type(s) of media used, along with analysis of comparative results (eg XLD vs BGA) would be useful.

Generally, the paper is well written and easy to read. I have made a few suggested minor edits to improve the English. There are also a couple of sentences that I found a little unclear and that need rewording or further clarification. (all edits etc are highlighted in the attached doc)

Author Response

Response to Reviewer 2 Comments, in addition the revised manuscript is attached

Point 1: I only have a few minor points that need addressing (see attached doc). The main point for consideration is in the use of alternatives to XLD. More information on the type(s) of media used, along with analysis of comparative results (eg XLD vs BGA) would be useful.

XLD..in 43%.. Is this XLD alone, and then XLD + another used in 86%? Sentence needs to be a bit more clear.

Do you have a breakdown of what the other plate types were, and if so are there comparative data on the results obtained with each type?

Response 1:                                                                             

  • The reviewer is right. More information on the use of alternatives to XLD could be useful. Moreover, the sentence about the use of XLD is not clear. For that reason we analysed the data on the use of the selective solid media further. A complete breakdown of all used media was prepared and put in supplemental Table S5.
  • In addition, there was a mistake in Figure 5. Plating was not described in 69 (86%) of the publications, but in 76 (95%). For this reason, a new version of Figure 5 was prepared. The figure is present in the revised manuscript attached.
  • The reviewer asks also for comparative data on the results obtained with each type of selective solid medium. This information cannot be given since these details are not in the different publications, moreover it was not the goal of this study to compare the different media.

Original sentence about selective plating-out (page 12 lines 381 and 382)

For selective plating-out, XLD was used in 34 (43%) of the studies and the use of a second agar was reported in 69 (86%) studies.

Proposal for replacement sentences (the sentence was replaced in the  revised manuscript attached).

For selective plating-out, XLD was used in 34 (43%) of the studies either alone (n=4) or in combination with one or more second selective solid media (n=30). As second selective solid medium to XLD, most frequently Brilliant Green agar (BGA) was used (n=9), followed by BrillianceTM Salmonella Agar (BSA) (n=8), MacConkey Agar (MCA) (n=7), RambachTM Agar (n=5), Hektoen Enteric Agar (n=5) and one or two times in five other selective solid media. XLD was not used in 46 (57%) of the studies. In 42 of these studies one or more other selective solid media as XLD were used whereas in 4 studies no use of such a medium was reported. In total, the use of 17 selective solid media was described in 76 (95%) studies. A complete breakdown of these media is in supplemental Table S5.

Point 2: Generally, the paper is well written and easy to read. I have made a few suggested minor edits to improve the English. There are also a couple of sentences that I found a little unclear and that need rewording or further clarification. (all edits etc are highlighted in the attached doc)

Response 2:

The text was edited and indicated sentences by the reviewer were reformulated. The edited sentences are marked in tract changes in the attached version of the revised manuscript.
